# GMM-Based Expanded Feature Space as a Way to Extract Useful Information for Rare Cell Subtypes Identification in Single-Cell Mass Cytometry

**DOI:** 10.3390/ijms241814033

**Published:** 2023-09-13

**Authors:** Aleksandra Suwalska, Joanna Polanska

**Affiliations:** Department of Data Science and Engineering, Silesian University of Technology, 44-100 Gliwice, Poland; aleksandra.suwalska@polsl.pl

**Keywords:** mass cytometry, identification of cell subpopulations, expanded feature space, Gaussian Mixture Models

## Abstract

Cell subtype identification from mass cytometry data presents a persisting challenge, particularly when dealing with millions of cells. Current solutions are consistently under development, however, their accuracy and sensitivity remain limited, particularly in rare cell-type detection due to frequent downsampling. Additionally, they often lack the capability to analyze large data sets. To overcome these limitations, a new method was suggested to define an extended feature space. When combined with the robust clustering algorithm for big data, it results in more efficient cell clustering. Each marker’s intensity distribution is presented as a mixture of normal distributions (Gaussian Mixture Model, GMM), and the expanded space is created by spanning over all obtained GMM components. The projection of the initial flow cytometry marker domain into the expanded space employs GMM-based membership functions. An evaluation conducted on three established cellular identification algorithms (FlowSOM, ClusterX, and PARC) utilizing the most substantial publicly available annotated dataset by Samusik et al. demonstrated the superior performance of the suggested approach in comparison to the standard. Although our approach identified 20 cell clusters instead of the expected 24, their intra-cluster homogeneity and inter-cluster differences were superior to the 24-cluster FlowSOM-based solution.

## 1. Introduction

Single-cell technologies have undergone significant advances in recent years, driven by the development of new computational methods. These technological advances have paved the way for investigating the heterogeneity of cell populations, providing valuable insights into the underlying mechanisms behind various diseases. Knowledge of cell states, dynamics, and regulatory mechanisms holds immense value in inventing new treatments and improving existing ones. However, analyzing the data provided by high-throughput technologies is challenging [1], mainly due to high dimensionality, which usually requires high-performance computing clusters and adjusted algorithms. Another challenge arises from evaluating solutions using reference annotations proposed by experts. Manual labeling introduces bias, potentially impacting the reliability of the results. Furthermore, the lack of consensus on rare cell subpopulations and their natural significance, especially in further subdivisions, adds complexity to the analysis [1,2,3]. As many biological functions and cell compositions remain not fully understood, careful interpretation and cautious exploration of the data is essential to advance our understanding of cellular processes and disease mechanisms [4].

Mass Cytometry by Time-of-Flight (CyTOF) is one of the single-cell technologies. In opposite to flow cytometry, it uses rare stable isotopes to label the antibodies for cell type identification and can measure from tens to dozens of markers simultaneously. Data from mass cytometry may contain millions of cells, making the analysis difficult [5].

The main objective in mass cytometry data analysis is to identify cell subpopulations after performing critical preprocessing steps like bead normalization, debarcoding, pre-gating, and batch effect correction [6]. Although several techniques have been proposed for each analysis step, each with specific requirements and conditions, a critical need remains for efficient solutions that can handle datasets containing millions of observations with the utmost precision [7]. Overcoming this challenge is essential to deepening our understanding of cellular heterogeneity and gaining valuable insights into the complexities of various diseases.

One of the major challenges faced by cell subpopulation identification methods is their limited sensitivity to rare cell types. Existing solutions often rely on manual expert annotations for evaluation, where algorithms are compared against these annotations. Manual annotations are typically based on selected markers that are characteristic of specific cell types, overlooking other relevant relationships in the feature space. This practice can introduce bias into the manual annotations. Consequently, solutions that do not perfectly match the known cell labels are often perceived as inferior, leading to a tendency to underrate the results.

Despite the availability of numerous methods for identifying cell populations, none can claim to be without limitations. The main challenge lies in scalability, as most methods struggle to handle large datasets containing millions of cells. For instance, methods that compute pairwise distances between observations often lead to a considerable matrix size that may exceed memory constraints, making it unfeasible to execute such computations. Therefore, some of the methods use downsampling, which may lead to rare subpopulations being overlooked.

Identifying rare populations is another challenge. While the large cell populations with easily distinguishable marker profiles are relatively easy to identify, the rare ones may have marker profiles too weak to be noticed. For example, they may express marker values in the middle of the marker’s range, making the information harder to extract. The existing solutions for identification often use dimensionality reduction techniques, like t-SNE [8,9], which may additionally influence the sensitivity of finding rare cell types.

In this study, a proposed modification of the feature space through its expansion is designed to help overcome the problem of low sensitivity in finding well-defined and separated groups of cells, including the rare ones. The idea is based on the assumption that the marker profile of one cell may overlap with another type, making it difficult to separate them into two clusters. Additionally, if the situation applies to the rare cell subpopulation, the group of cells is almost impossible to identify. Since the marker’s distribution is a mixture of Gaussians, each Gaussian potentially describes a different set of characteristic observations. Marker values can be assigned to the appropriate component of the Gaussian Mixture Model by calculating the conditional probability. Each cell then has a set of conditional probabilities that determine the components that most characterize it. Consequently, the calculated conditional probabilities create new features with values bounded to a range [0, 1], which also acts as a normalization technique for the machine learning algorithms. The idea is visualized in Appendix A.

## 2. Related Work

Some clustering techniques are dedicated to identifying cell subpopulations in mass cytometry data. They usually adapt classical machine learning clustering techniques with data preprocessing steps, such as hierarchical clustering, combined with density-based downsampling of the observations. One of the best methods [3] in terms of detection sensitivity, stability of the results, and time of computation is FlowSOM [10]. Other solutions are those dedicated to different biological datasets, including mass cytometry, like PARC [11] or ClusterX [12].

FlowSOM [10] is based on Self-Organizing Map (SOM). The algorithm builds and trains SOM and connects the resulting nodes in a minimal spanning tree. The meta-clustering step with a known number of clusters gives the final result. PARC [11] constructs a nearest-neighbor graph with a hierarchical navigable small world. Next, it prunes the graph’s edges based on the distribution of weights. PARC uses the Leiden algorithm to find communities that determine cell types. The algorithm works fast and accurately for high-dimensional data, effectively finding rare cell subpopulations.

The SPADE method [13] comprises four primary modules: data downsampling, clustering, creation of a minimal spanning tree to connect the clusters, and upsampling, where the remaining cells are mapped into the identified groups. This method requires four input parameters: markers for building the tree, which necessitates prior knowledge of potentially informative features; the density of outliers and targets governing the downsampling process; and the number of clusters.

In contrast, the DensVM [8] approach employs t-SNE [14] dimensionality reduction and the resulting embedding is fed into clustering and 2D peak-finding algorithms. Each peak represents a cluster centroid, to which cells are assigned based on the smallest distance. The algorithm’s cluster assignments serve as labels in the Support Vector Machine (SVM) model for classification. Any cells that remain unlabeled by the method are subsequently employed as a test set during SVM training.

Another algorithm is based on k-Nearest-Neighbor (k-NN): ClusterX [12] reduces the dimensionality with t-SNE and measures for each point the local density and the distance to points with higher local density. As a result, the centroids are determined, and the rest of the points are assigned to the closest one creating a cluster.

PhenoGraph [15] employs a nearest-neighbor graph (k-NNG) to represent cells as nodes connected to their most similar counterparts. Subsequently, community detection is utilized to identify phenotypically similar cell subpopulations. ImmunoClust [16], on the other hand, adopts Finite Mixture Models to cluster cell events initially. Then, cell clusters are grouped and merged across samples (meta-clustering) to yield the final results.

X-Shift [9] is an algorithm based on k-NN density estimation. A density estimate is computed for each data point and connections are established to the nearest neighbor with a higher density estimate. Points without such neighbors are considered potential cluster centroids. The algorithm further iteratively connects centroids and merges clusters based on Mahalanobis distance until a predefined condition is fulfilled.

## 3. Results

The publicly available dataset [9] was preprocessed and visualized as a mISO plot [17] in two-dimensional space (Figure 1A). The appropriate cell types, identified by the experts, are indicated in the visualization with different colors. The cell populations in the reduced space are placed in a way that preserves biological similarities between them.

After preprocessing (ArcSinh transformation, batch effect correction with cyCombine), the original feature space was expanded with the proposed algorithm. Each marker’s intensity values were decomposed with Gaussian Mixture Model and Bayesian Information Criterion (BIC) to find the optimal number of mixture components. The number of components found for the markers varied from 2 to 13.

For each GMM, conditional probability values were calculated and visualized with a line plot. The existing artifacts were then corrected, resulting in the membership function. Visualization of the effect of the correction is presented in Appendix A. As can be observed, the correction effectively removed inactive components, corrected lines that dominate in more than one range of marker expression values and corrected the order of the components. Then, the membership values to each new feature (component) were calculated for each cell. Figure 2 depicts five exemplary cell subpopulations and their highest expression of a given marker to present how the expanded feature space influenced the cell type characteristics. For example, NKT cells are characterized by deficient expression of B220 and IgM markers, very high expression of the CD45 marker, and a medium expression of the CD44 marker. In the new expanded feature space, it means that the NKT cells will have high values, mainly for the first component of B220 (the first feature created from the marker).

The final number of features increased from 38 (regular domain) to 318 (expanded domain). For the expanded feature space, the mISO plot with expert annotations (ground truth) was also created (Figure 1B). An interesting observation is that the cell types belonging to one family of cells lay closer in the expanded feature space. The separation between cell groups is more extensive than in the regular feature space (Figure 1C,D).

The dataset in the regular and expanded feature space was applied to clustering with three chosen algorithms: FlowSOM, ClusterX, and PARC. The results are presented in Figure 3 for the regular space and Figure 4 for the expanded space.

The evaluation metrics, Calinski-Harabasz (CHI) and Davies–Bouldin Indices (DBI), are collected in Table 1. After the expansion of the feature space, the Calinski–Harabasz Index was higher and the Davies–Bouldin Index was lower, which indicates an improvement in the definition and separation of the resulting clusters. However, the number of clusters (determined automatically for ClusterX and PARC) was lower than in the regular space and the number set by experts. The best results were obtained for the PARC clustering algorithm in the expanded feature space.

## 4. Discussion

The analysis of single-cell mass cytometry data is a challenging task, that requires thorough investigation and methodological implementation to account for the high-dimensional feature space and data heterogeneity. The presence of data heterogeneity can have a significant impact on clustering results, potentially leading algorithms to discover a larger number of cell groups than those identified by experts. Effective identification of rare cell populations is hindered by the complexity introduced by high-dimensional data, especially when there is a notable discrepancy in the cell counts between specific and rare cell groups. These challenges remain an active area of research as many researchers strive to find the most optimal solution to overcome these intricacies.

Although many clustering algorithms are dedicated to mass cytometry [8,9,10,11,12,13,14,15,16], none are perfect. Usually, they focus on the algorithm itself rather than the modification of the feature space, which may also be beneficial. Typically, high-dimensional datasets are subjected to downsampling [13,15,16] and dimensionality reduction techniques [8] to eliminate non-informative features rather than expanding dimensions. However, in the context of expanding dimensions, the new features encapsulate a subset of information derived from the original marker expression values. It is hypothesized that these new features are inherently more informative than the original ones and can significantly aid in the identification of even rare cell populations. Therefore, in this study, we have proposed such a modification of the feature space to help the algorithm extract useful information for the identification process.

The study uses a publicly available dataset from Samusik et al. (2016) [9], which is the largest annotated mass cytometry dataset, to the best of our knowledge. After preprocessing, the feature space was expanded and corrected, resulting in a noteworthy increase from 38 to 318 features. The proposed correction algorithm for conditional probability lines effectively achieves its intended goals, including removing inactive components and establishing the correct order and single ‘peak’ for each line. However, the algorithm has certain limitations and disadvantages. For datasets comprising a substantial number of observations, the Gaussian Mixture Decomposition process may become time-consuming and produce a large number of components. To mitigate these challenges, it is necessary to adapt the parameters and criteria of the algorithm, such as the minimal allowed sigma for Gaussian components.

The transformed lines no longer represent conditional probabilities, leading to the adoption of the term ‘membership function’. Nevertheless, it is essential to note that the lines retain their original shape at the peaks’ summits and the points of intersection, making them still applicable for determining cut-off values—a prevalent practice in diverse domains. The algorithm will undergo several improvements to enhance efficiency and achieve superior results, selectively targeting lines that necessitate correction. Rigorous testing of the correction algorithm has been carried out on diverse datasets and its availability to the public is assured, with regular updates being provided.

Among the various methods proposed for cell type identification thus far, only a few can handle large datasets and effectively identify rare subpopulations. To investigate their performance, this study compared the three best methods using Samusik’s dataset in both the regular and expanded feature space domains. Notably, the clustering of the dataset in the expanded domain yielded better results than in the regular domain, confirming the usefulness of dimensionality expansion. The best results were obtained for the PARC algorithm (both domains), with the best values of CHI = 95,594.57 and DBI = 1.3168 for the expanded feature space domain. Most manual clusters, including the rare ones, were identified, which is 20 out of 24 for PARC.

The introduction of an expanded feature space may potentially have a negative impact on the clustering results, primarily due to the phenomenon known as the curse of dimensionality. Although the new features carry valuable information capable of distinguishing various cell types, the model may struggle to use this information effectively when making clustering decisions based on all available features. Consequently, the use of the expanded domain may result in fewer cell populations being identified. Additionally, certain algorithms may not perform optimally when feature distributions deviate significantly from Gaussian distributions. The expanded feature space includes observations at two ends of the feature range: values near zero for cells not expressing the particular feature and values near one for cells expressing the feature.

The distribution of these values differs significantly from the original markers’ distributions, as shown in Appendix A. As a result, traditional identification approaches are expected to perform sub-optimally. To exploit the full potential of the expanded feature space, algorithms need to be adapted and tailored to suit the unique characteristics of the domain. Ongoing efforts involve the development of feature selection and cell-type identification algorithms explicitly designed to operate effectively within the proposed expanded feature space.

This study addressed the challenge of inefficient rare cell subpopulation identification in large mass cytometry data. To overcome this limitation, we proposed to modify the feature space to extract meaningful information that can aid cell type identification. The proposed expanded feature space comprises GMM components as separate features derived from the decomposition of each marker’s expression values. Remarkably, when existing techniques were applied using the expanded feature space, superior results were obtained compared to the regular space, underscoring its intrinsic value. Most manual clusters, including the rare ones, were identified. Our approach represents a novel contribution to the field, demonstrating significant potential. However, the existing solutions for determining cell types are not optimized to handle such a specific feature space. Therefore, new methods and feature selection algorithms need to be developed to achieve an optimal solution.

## 5. Materials and Methods

### 5.1. Dataset

The publicly available dataset from Samusik et al. [9] was used in the study. The dataset consists of 514,386 annotated cells and 38 markers categorized into 24 populations. The cells were annotated by three experts in the manual gating process. The cells came from 10 mouse samples.

### 5.2. Preprocessing and Visualization

The dataset was preprocessed (ArcSinh transformation with a co-factor of 5) and corrected for batch effect with cyCombine [18].

A mISO plot [17] was generated to understand the data better. The mISO plot is a proposition of visualizing high-dimensional labeled datasets in the two-dimensional UMAP projection. The method presents the clusters (cell types) as regions with the highest concentration of observations instead of coloring each point, which makes the visualization unclear.

### 5.3. Artifacts Associated with the Expanded Feature Space

In order to expand the feature space, the dataset underwent processing using Gaussian Mixture Models (GMMs) for each marker. Selecting the most suitable number of components was achieved through the Expectation Maximization algorithm, supplemented by the Bayesian Information Criterion [19]. Utilizing the resulting GMMs, conditional probabilities were computed for each marker (Appendix A). Subsequently, the expression values of cells were replaced with their corresponding conditional probabilities, resulting in the generation of novel features. This feature space expansion strategy augments the dataset’s descriptive power, leveraging probabilistic insights for improved performance in various analytical and machine-learning tasks.

Certain artifacts must be corrected for the results to be reliable and to enable the effective use of conditional probabilities as new features. Each Gaussian Mixture Model component is characterized by a specific set of parameters, and an expression value holds different probabilities of belonging to each component. Notably, the sum of probabilities for any expression value always amounts to 1. Visualizing these probabilities as lines makes the data more interpretable and analyzable. The visualization is similar to the probability density function for a given GMM, with some differences. The first and last of the conditional probability lines should start and end at a level equal to one, respectively, since the points at both ends of the marker distribution have the smallest distance to the first and last of the components.

Due to the calculation of conditional probability lines from GMM components, certain artifacts may arise as a result of the inherent nature of GMMs. When two Gaussians partially overlap, determined by their mean and standard deviation values, the Gaussian with a higher standard deviation will exhibit a broader shape and, eventually, higher values on its ends compared to the other Gaussian. This behavior can cause one conditional line to have two “peaks”, leading it to dominate in more than one range of expression values. Consequently, this may also alter the order of the components. Furthermore, inactive components that fail to dominate in any range of values pose another challenge. Figure 5 illustrates these artifacts.

Suppose there exist conditional probability lines dominating multiple segments of the *x*-axis. In that case, it may lead to cells with similar marker values being assigned to different Gaussian Mixture Model components while another cell displaying a more substantial difference in the expression value is included in the same component. In the presented case in Figure 5B, a cell with a marker value of x1 is part of the same GMM component as the cell expressing x3 of the marker, while a cell close to the marker value x2 is allocated to a distinct GMM component.

Figure 5A depicts the source of the dominance of a component in different ranges of values on the example of the CD45 marker from Samusik’s dataset. The image shows the zoomed area at the beginning of the marker’s expression values distribution. It is evident that the first peak (depicted in dark red, left panel), corresponding to the first component with the smallest mean value, is predominantly influenced by the light red and violet lines (right panel). Consequently, expression values within the range of approximately [0, 0.125] are more likely to belong to the light red or violet components, representing Gaussian distributions with higher mean values than the nearest component, represented by the dark red. The same problem may occur at the end of the marker distribution.

In Figure 5C, the conditional probability lines plot reveals the presence of an inactive component in the model. The red arrow is used to pinpoint this inactive component within the Gaussian Mixture Model. Due to its significantly small weight, the component lacks the ability to exert any substantial influence across the range of marker values. As it does not add any value to the expanded feature space, the logical step is to exclude this component from the set of components.

### 5.4. Correction of Artifacts and Membership Function

To establish an appropriate expanded feature space, a series of steps must be implemented as follows:Correct the conditional probability lines to exhibit only one maximum (a single peak) throughout the entire range of marker values.Identify the first and last components’ lines and restore their dominance at the beginning and end of the distribution.Ensure that the first and last conditional probability lines commence and terminate (respectively) at a probability value of 1, resulting in a distinct shape modeled differently from the rest of the lines.Identify components with excessively low component proportion (weight), rendering them non-dominant in any marker value range. Consequently, remove these components from the expanded feature space.While correcting the conditional probability lines, preserving the points of intersection among the verified lines is crucial. These intersections often serve as thresholds in various applications. As a result, meticulous modeling of the lines above the intersection points is essential to reflect their original shape accurately.

The final new features after correction represent a membership function. The correction steps are visualized in Figure 6.

The correction algorithm implemented in MATLAB2022b satisfactorily fulfills all the above-prescribed criteria, operating directly on the conditional probability lines and their shapes. Nonetheless, due to the modification of the lines during the correction process, the resulting probabilities for each *x*-axis value no longer sum up to 1, thereby rendering the term ‘conditional probability’ inappropriate. In light of this, we suggest adopting the term ‘membership function’, as the values now signify the membership of observations to the respective components. The main parts of the pipeline are presented in Figure 6.

Initially, the algorithm verifies the presence of a peak near the mean value parameter for each component’s line, signifying its authenticity. Subsequently, it examines whether each component has a distinct range of values where it exhibits dominance over the other lines. If a component is deemed inactive, it is eliminated from the model. Following this removal, the probabilities are recalculated to ensure they sum up to 1.

In the following phase, the algorithm focuses on correcting the line associated with the first component. The correction process involves transforming the line’s shape to commence from the point (0, 1), representing the maximum membership value for the marker expression of 0. To accurately depict the line’s behavior for higher expression values, the algorithm samples a set of points from the right side of the peak. Eventually, the line concludes with a membership value of 0, which aligns with the maximum expression value. Upon achieving the necessary improvements to the first component, a similar strategy is implemented to model the middle components. However, during this process, the lines’ initiation and termination points are both set to a membership level of 0. The last component is then subjected to correction, ensuring it concludes with a membership level of 1 for the maximum marker value. The lines are effectively modeled by employing the ‘pchip’ function on a set of points within the interval of m ± 3 s (where m denotes the mean value and s represents the standard deviation) and the maximum and minimum values of the entire expression range.

The resulting membership function assigns each *x*-axis value (marker expression) to the closest component, indicated by the highest membership value, and to neighboring components with smaller membership values.

### 5.5. Clustering

The dataset in the regular and expanded feature spaces was applied to clustering to identify cell subpopulations with the three best methods: FlowSOM, ClusterX, and PARC. The algorithms were chosen because of their adaptation to millions of cells, fast computation, and high accuracy compared to other methods. The algorithms were run with the default parameters. Because FlowSOM requires knowledge about the final number of clusters, it was set to 24 (as the number of cell types identified by experts). The results were then compared in terms of the Calinski–Harabasz Index [20], Davies–Bouldin Index [21], and the number of clusters found (ClusterX and PARC).

### 5.6. Technical Details

The analysis was conducted using MATLAB R2022b, Python 3.9, and R 4.2.2 programming environments. The calculations were carried out on GeCONiI servers (AMD CPU 256 threads, 2.6G, 2TB RAM).

## Figures and Tables

**Figure 1 ijms-24-14033-f001:**
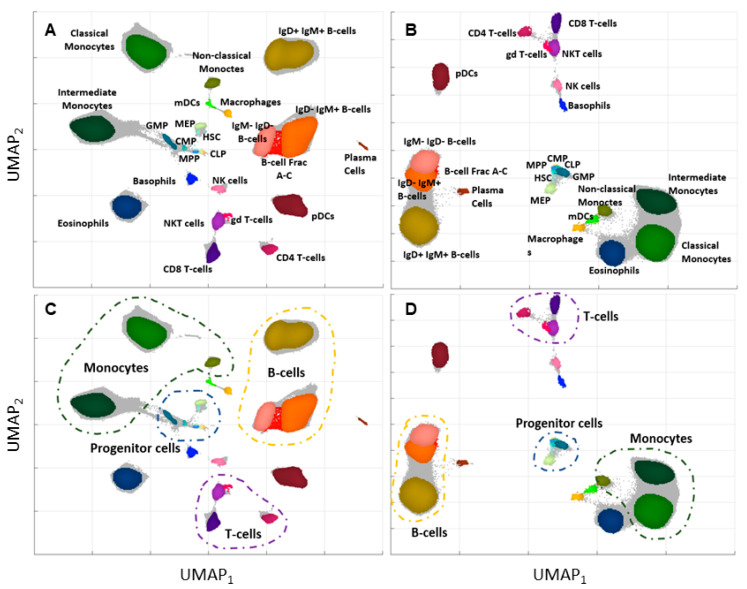
mISO plot of the regular and expanded feature spaces highlights how they affect the positioning of cell types. (**A**) Depicts the regular feature space, annotated by domain experts, while (**B**) displays the expanded feature space with the appropriate annotations provided by experts. The positions of cell groups vary in the expanded feature space compared to the regular space while preserving the similarities between cell types. Further insights are provided in (**C**,**D**), unveiling the primary groups of similar cell types in the regular and expanded domains, respectively.

**Figure 2 ijms-24-14033-f002:**
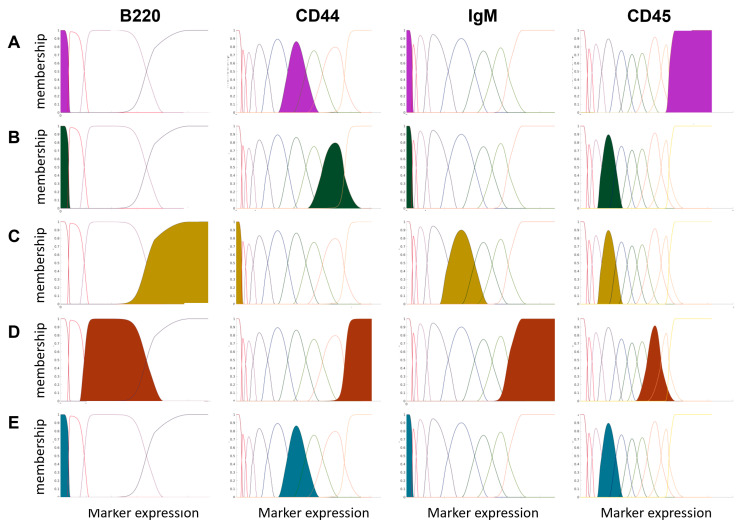
A comparison of marker profiles for five selected cell types in the expanded domain. The colored components signify expression levels for each cell type based on the highest membership value, determined by the maximum value of the 90th percentile for each component. Given that each component contributes to a new feature in the expanded domain, they provide valuable insights into distinct cell populations. (**A**) NKT cells; (**B**) intermediate Monocytes; (**C**) IgD+ IgD+ B cells; (**D**) plasma cells; (**E**) GMP.

**Figure 3 ijms-24-14033-f003:**
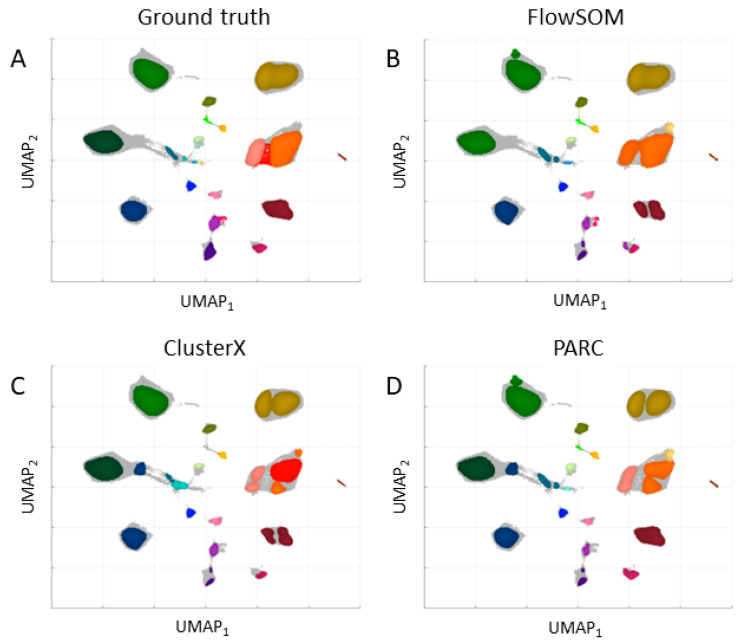
Results for the three existing cell-type identification approaches for a dataset in the regular feature space. (**A**) Known assignments provided by experts; (**B**) 24 clusters identified by FlowSOM; (**C**) 25 clusters identified by CusterX; (**D**) 24 clusters identified by PARC.

**Figure 4 ijms-24-14033-f004:**
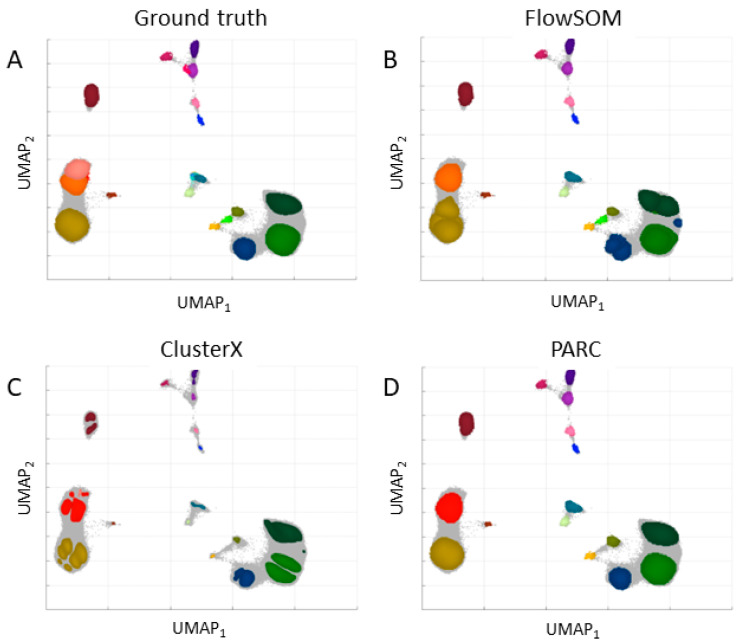
Results for the three existing cell-type identification approaches for a dataset in the expanded feature space. (**A**) Known assignments provided by experts; (**B**) 24 clusters identified by FlowSOM; (**C**) 19 clusters identified by CusterX; (**D**) 20 clusters identified by PARC.

**Figure 5 ijms-24-14033-f005:**
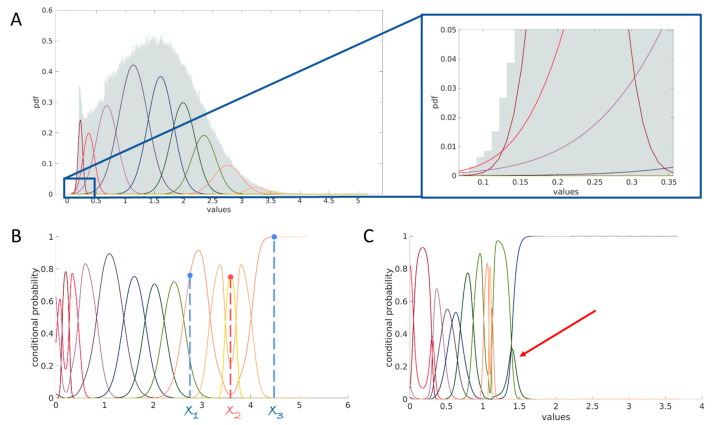
The primary issues associated with using conditional probabilities for assigning cells to Gaussian Mixture Model (GMM) components. (**A**) Demonstrates the occurrence of the dominance of a latter component before the first one, leading to a change in the order of component dominance and/or the presence of multiple peaks for a single component. Such scenarios can also be observed at the end of the distribution. (**B**) Illustrates the impact of having two dominance peaks for one or more conditional lines. As depicted, cells with marker expressions x1 and x3 exhibit the highest probability of belonging to the same component, while the cell with x2 expression, which is more similar to x1 and x3, belongs to another component. (**C**) Presents an example of an inactive component that lacks any form of dominance.

**Figure 6 ijms-24-14033-f006:**
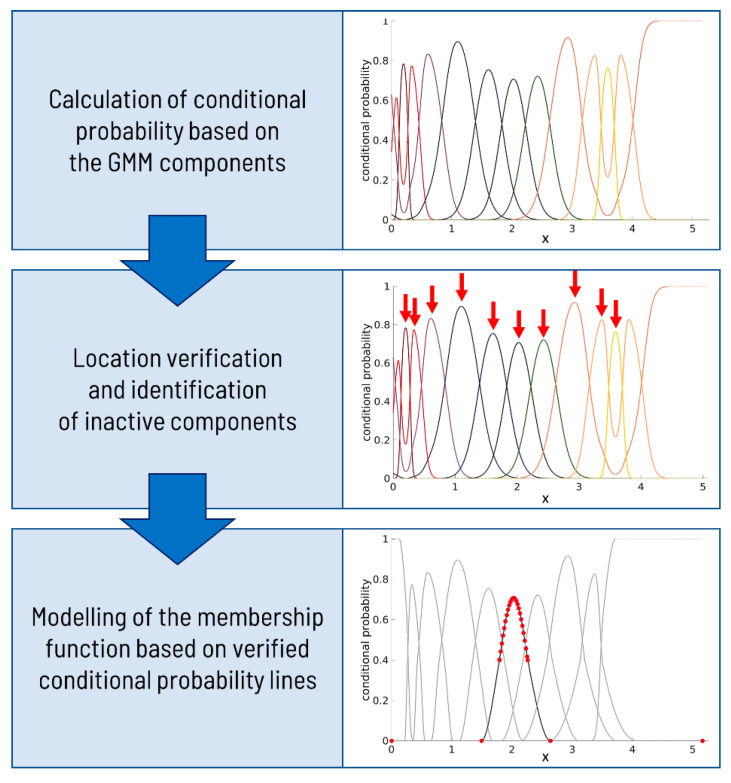
The pipeline of conditional probability lines correction. Initially, a Gaussian Mixture Model (GMM) decomposition is carried out for a given marker, yielding a set of mixture components. Subsequently, conditional probabilities are calculated based on these components, visualized as a series of lines. Each line represents the probability associated with a specific marker value, determining its assignment to a corresponding component. The peaks in the lines signify regions where the respective component exhibits dominance, indicating the range of marker values attributed to that particular component. The red arrows point to the correct dominance of the components that will be preserved. For each indicated peak, a set of points is chosen (red dots) to model a new shape: the peak is modeled accurately, but the edges are forced to fall to the zero value of conditional probability, preventing the occurrence of additional peaks.

**Table 1 ijms-24-14033-t001:** Comparison of the results from chosen existing methods in regular and expanded domains.

	Regular Domain			Expanded Domain		
Algorithm	#Clusters	CHI	DBI	#Clusters	CHI	DBI
FlowSOM	24	73,251.44	1.4552	24	88,863.82	1.3356
ClusterX	25	78,742.18	1.4573	19	90,746.60	1.3944
PARC	24	82,305.73	1.4162	20	95,594.57	1.3168

## Data Availability

The dataset used in the study is available under [9]. The correction algorithm is published under the link: https://github.com/Aleksandra795/membership_function.git (accessed on 13 March 2023).

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
