# Peer review of "GMM-Based Expanded Feature Space as a Way to Extract Useful Information for Rare Cell Subtypes Identification in Single-Cell Mass Cytometry"

_ijms, 2023, doi:10.3390/ijms241814033_

Round 1

Reviewer 1 Report

The manuscript entitled "Expanded feature space – as a way to extract useful information for rare cell subtypes identification in single-cell mass cytometry." describes an algorithm for cytometry data based on Gaussian mixture model decomposition to expand the number of features. The expanded feature space was subsequently used by three different common clustering methods, FlowSom, ClusterX and PARC. 

How does this algorithm compare against other Gaussian Mixture Models based algorithm such as Immunoclust and Swift? What is the difference between this approach and the other GMM based approaches?

Only a single existing dataset is chosen, I suggest to consider at least 3 datasets for a fair comparison. 

The datasets chosen consisted of 24 annotated cell populations. Why was the expanded feature + PARC with only 20 found clusters better than the 24 clusters found by FlowSOM? Wouldn't you expect to find 24 clusters?

The references to the Figures/Tables are not present: (Error! Reference source not found..A) line 131, 146, 170, 173 176 and 179. 

Author Response

  1. How does this algorithm compare against other Gaussian Mixture Models based algorithm such as Immunoclust and Swift? What is the difference between this approach and the other GMM based approaches?

Answer: Gaussian Mixture Models (GMM) have been utilized since 1894, when Karl Pearson devised the method to mathematically represent the heterogeneous distribution of crabs in the Bay of Naples. GMM is predominantly employed as a tool for feature engineering, such as unsupervised feature selection/filtering, and/or data clustering/classification. Our approach uses Gaussian mixture models in the feature engineering phase, which sets it apart from the ImmunoClust and Swift algorithms that were referenced by the reviewer. Unlike those methods, which employ GMM for cell clustering, our technique leverages GMM to portray the intensity distribution of each Mass Cytometry marker. This enables us to project the data into a new feature space with enhanced dimensions. Essentially, this expansion enables different expected marker values with specific variance (or combination of few of them) to describe each cell subpopulation. The projection of the initial feature space into the expanded space, derived from the data, improves separation among the subpopulations of cells, increasing the ability to identify rare cells during clustering methods applied to the expanded version. As a feature engineering technique, our methods allow for the use of nearly any clustering algorithm, including those based on GMM, for identifying cell subpopulations. This simplifies the task.

Additionally, ImmunoClust uses the finite mixture model to cluster cells and the resulting clusters’ centroids. However, the clustering is performed in the original feature space (marker values). The Swift algorithm decomposes each marker with GMM. However, this step is performed as a part of the clustering algorithm that operates in all dimensions simultaneously. The resulting components (potential subpopulations) become clusters based on the splitting and merging criterion depending on the multimodality. The clusters are determined using sampling of observations from the dataset and are split with the Expectation-Maximization algorithm until all are unimodal. The steps are repeated for other samples until all cells are considered. Then, the clusters are merged if the resulting one is unimodal in each dimension. Moreover, the Swift algorithm is intended for flow cytometry datasets with limited markers (usually around 8-15). In that dimensionality, the algorithm may work well, but in mass cytometry, it may be challenging to find unimodal clusters with GMM among 38 dimensions (the number of markers in the dataset used in the study). It is known that solutions dedicated to flow cytometry perform worse on mass cytometry data.

To highlight distinctions between our methodology and others, we altered the title of the manuscript to: GMM-based expanded feature space for extracting valuable insights to identify rare cell subtypes in single-cell mass cytometry.

  1. Only a single existing dataset is chosen. I suggest to consider at least 3 datasets for a fair comparison.

Answer: We fully understand the reviewer's concern about the lack of validation of the methods on multiple independent datasets. There are several publicly available mass cytometry datasets (examples cited below) and they vary in size (number of cells and markers) and whether annotations are provided. Most of them limit the annotations to the main cell types and/or do not include the rare subpopulations. In addition, the number of cells is usually less than tens of millions. Our team collaborates with biologists from Stellenbosch University, South Africa, and based on their expert opinion and experimental needs, we focus on analysing datasets with tens or even hundreds of millions of cells, as only these allow the detection of rare cell subpopulations. Therefore, we limit ourselves to developing algorithms to work on large datasets, but the largest annotated dataset we could obtain, to the best of our knowledge, is the Samusik dataset used in our study. Despite an intensive literature/database search, we could not find any other similar, well annotated dataset. We would be grateful for any links/suggestions to datasets that we can use for our purposes.  

Examples of the publically available mass cytometry datasets that can be downloaded from flowrepository.org:

[1] Trussart M, et al., Removing unwanted variation with CytofRUV to integrate multiple CyTOF datasets. Elife. 2020 Sep 7;9:e59630. doi: 10.7554/eLife.59630. PMID: 32894218; PMCID: PMC7500954. http://flowrepository.org/public_experiment_representations/2722

[2] Simoni Y, et al., Human Innate Lymphoid Cell Subsets Possess Tissue-Type Based Heterogeneity in Phenotype and Frequency. Immunity. 2017 Jan 17;46(1):148-161. doi: 10.1016/j.immuni.2016.11.005. Epub 2016 Dec 13. Erratum in: Immunity. 2018 May 15;48(5):1060. PMID: 27986455; PMCID: PMC7612935. https://flowrepository.org/id/FR-FCM-ZYZX

[3] Bendall SC, et al., Single-cell mass cytometry of differential immune and drug responses across a human hematopoietic continuum. Science. 2011 May 6;332(6030):687-96. doi: 10.1126/science.1198704. PMID: 21551058; PMCID: PMC3273988. https://flowrepository.org/id/FR-FCM-ZY9R

  1. The datasets chosen considered of 24 annotated cell populations. Why was the expanded feature + PARC with only 20 found clusters better than the 24 clusters found by FlowSOM? Wouldn’t you expect to find 24 clusters?

Answer: The two methods diverge regarding their pipeline. FlowSOM requires a predetermined number of clusters, and there is no dependable method of automatically estimating it. After considering Samusik's annotation data, we opted for the most unbiased alternative and set the cluster count at 24. On the other hand, PARC automatically recognizes the necessary number of clusters and generated a solution consisting of 20 clusters. However, 24 FlowSOM clusters - although their number agreed with expert opinion - were less separated and less homogeneous (Calinski-Harabasz=88863.82, Davies-Bouldin=1.3356) when compared to the 20 clusters identified by PARC (Calinski-Harabasz=95594.57, Davies-Bouldin=1.3163). In addition, upon analysis of the FlowSOM-based clustering results, it is also observed that the 24 clusters identified by FlowSOM do not fully correspond to the actual cell notations provided by the experts. Instead, FlowSOM divided larger clusters of the same cell type into smaller ones, disregarding other (including rare) subpopulations of cells.

  1. The references to the Figures/Tables are not present: (Error! Reference source not found...) line 131, 146, 170, 173, 176 and 179.

Answer: Thank you for the comment. We did not notice that there are problems with references to the figures. The issue has been fixed.

Reviewer 2 Report

Date:

Journal: International journal of Molecular Science

Title: Expanded feature space – as a way to extract useful information for rare cell subtypes identification in single-cell mass cytometry.

Dear Editor,   

In this manuscript, the authors propose a modification of the feature space through its expansion to help overcome the problem of low sensitivity of finding well-defined and separated groups of cells, including the rare ones.

The manuscript is recommended to be accepted after minor revision.

Originality: moderate

Clarity of Presentation: moderate

Importance to Field: moderate

Comments for authors:

1.     The abstract is very short and has some very long paragraphs. It should introduce more details. Some corrections are highlighted in pdf file

2.     Keywords: Some corrections are highlighted in pdf file

3.     The introduction has some very long paragraphs a is very long. Also, authors should unite all sentences either in the past tense or in present tense.   Some corrections are highlighted in pdf file

4.     Related work: add section of related work after introduction with references

5.     Results: Some corrections are highlighted in pdf file

6.     Discussion: Some corrections are highlighted in pdf file

7.      English editing is recommended.

8.      References: must be adjusted according to style of the journal ,authors should add more references as the number is small

9.      Graphical Abstract: I recommend to add graphical abstract summarize the whole idea of paper.

10.  Highlights: authors should add  

11.  The table and figures should be improved

Generally, the manuscript needs to revise well especially for typing, grammatical, and

formatting mistakes, requires careful linguistic revision by native English speakers

-Please revise the abbreviations in the whole manuscript and write the full name

The methods section is well-written and provides a clear description of the study

design, participant selection, and genotyping methods. However, the following should

be considered:

- providing the IRB's ethical approval number

Author Response

  1. The abstract is very short and has some very long paragraphs. It should introduce more details. Some corrections are highlighted in pdf file

Answer: Thank you for the comment; the abstract has been modified. The corrections highlighted in pdf file were included in the manuscript.

  1. Keywords: Some corrections are highlighted in pdf file.

Answer: The corrections highlighted in pdf file were included in the manuscript.

  1. The introduction has some very long paragraphs a is very long. Also, authors should unite all sentences either in the past tense or in present tense. Some corrections are highlighted in pdf file

Answer: Thank you for the comment. We adjusted the introduction. The corrections highlighted in pdf file were included in the manuscript.

  1. Related work: add section of related work after introduction with references

Answer: Thank you for the suggestion. The section has been added to the manuscript.

  1. Results: Some corrections are highlighted in pdf file

Answer: The corrections highlighted in pdf file were included in the manuscript.

  1. Discussion: Some corrections are highlighted in pdf file

Answer: The corrections highlighted in pdf file were included in the manuscript.

  1. English editing is recommended

Answer: The English was improved.

  1. References: must be adjusted according to style of the journal, authors should add more references as the number is small

Answer: The formatting of references has been adjusted to the style of the journal and we added more references.

  1. Graphical Abstract: I recommend to add graphical abstract to summarize the whole idea of paper

Answer: Thank you for the suggestion, we have prepared the graphical abstract.

  1. Highlights: authors should add

Answer: Thank you for the comment. We have added the highlights.

  1. The table and figures should be improved

Answer: We have improved the figures.

  1. Please revise the abbreviations in the whole manuscript and write the full name

Answer: The abbreviations have been revised.

  1. The methods section is well-written and provides a clear description of the study design, participant selection, and genotyping methods. However, the following should be considered: - providing the IRB's ethical approval number

Answer: The dataset used in the study was publicly available [16]. We did not conduct any additional measurement experiments or collect new samples, so we did not need additional approval from the ethics committee.